

# Relationships between consumption of ultra-processed foods, gestational weight gain and neonatal outcomes in a sample of US pregnant women

Karthik W. Rohatgi[1], Rachel A. Tinius[2], W. Todd Cade[1],
Euridice Martínez Steele[3], Alison G. Cahill[4] and Diana C. Parra[1]

[1] Program in Physical Therapy, Washington University School of Medicine, St. Louis, MO,
United States of America
[2] School of Kinesiology, Recreation, and Sport, Western Kentucky University, Bowling Green, KY,
United States of America
[3] Department of Nutrition, University of São Paulo, São Paulo, Brazil
[4] Department of Obstetrics and Gynecology, Washington University School of Medicine, St. Louis, MO,
United States of America

Corresponding author
Karthik W. Rohatgi,
karthik.rohatgi@wustl.edu

## ABSTRACT

**Background.** An increasingly large share of diet comes from ultra-processed foods (UPFs), which are assemblages of food substances designed to create durable, convenient and palatable ready-to-eat products. There is increasing evidence that high UPF consumption is indicative of poor diet and is associated with obesity and metabolic disorders. This study sought to examine the relationship between percent of energy intake from ultra-processed foods (PEI-UPF) during pregnancy and maternal gestational weight gain, maternal lipids and glycemia, and neonatal body composition. We also compared the PEI-UPF indicator against the US government's Healthy Eating Index-2010 (HEI-2010).

**Methods.** Data were used from a longitudinal study performed in 2013–2014 at the Women's Health Center and Obstetrics & Gynecology Clinic in St. Louis, MO, USA. Subjects were pregnant women in the normal and obese weight ranges, as well as their newborns ($n = 45$). PEI-UPF and the Healthy Eating Index-2010 (HEI-2010) were calculated for each subject from a one-month food frequency questionnaire (FFQ). Multiple regression (ANCOVA-like) analysis was used to analyze the relationship between PEI-UPF or HEI-2010 and various clinical outcomes. The ability of these dietary indices to predict clinical outcomes was also compared with the predictive abilities of total energy intake and total fat intake.

**Results.** An average of $54.4 \pm 13.2\%$ of energy intake was derived from UPFs. A 1%-point increase in PEI-UPF was associated with a 1.33 kg increase in gestational weight gain ($p = 0.016$). Similarly, a 1%-point increase in PEI-UPF was associated with a 0.22 mm increase in thigh skinfold ($p = 0.045$), 0.14 mm in subscapular skinfold ($p = 0.026$), and 0.62 percentage points of total body adiposity ($p = 0.037$) in the neonate.

**Discussion.** PEI-UPF (percent of energy intake from ultra-processed foods) was associated with and may be a useful predictor of increased gestational weight gain and neonatal body fat. PEI-UPF was a better predictor of all tested outcomes than either total energy or fat intake, and a better predictor of the three infant body fat measures

than HEI-2010. UPF consumption should be limited during pregnancy and diet quality should be maximized in order to improve maternal and neonatal health.

## INTRODUCTION

It has been well-documented that nutrition before and during pregnancy can have long lasting effects on maternal and neonatal health outcomes (*Imhoff-Kunsch & Martorell, 2012*). In particular, consumption of ample fruits, vegetables, whole grains, and lean meats, and limited consumption of caffeine, alcohol, and foods high in saturated fat during pregnancy has been recommended (*National Health and Medical Research Council (NHMRC), 2013*; *National Health Service (NHS), 2017*). Evidence has emerged showing that consumption of foods high in sugar (*Petherick, Goran & Wright, 2014*), saturated fat (*Park et al., 2013*) and sodium during pregnancy can be particularly harmful to both the pregnant woman and their neonates (*Tay et al., 2012*). Many of these foods can be categorized as ultra-processed foods (UPF), which are assemblages of food substances designed to create durable, accessible, convenient and palatable ready-to-eat or ready-to-heat food products (*Monteiro et al., 2017*). These products are often consumed as snacks instead of home-prepared dishes, are low in fiber, whole grains, and vitamins (*Monteiro et al., 2017*) and include artificial colors, flavors, and preservatives, which can be particularly harmful for pregnant women (*Halldorsson et al., 2010*).

Ultra-processed foods (UPFs) are merely one group in a four-category classification system (NOVA) that was developed to guide consumers towards a healthy diet using food-based, rather than nutrient-based, dietary guidelines (*Monteiro et al., 2017*). There is increasing evidence that high consumption of UPFs is indicative of poor diet and is associated with obesity, metabolic syndrome and cardiovascular disease in non-gravid adults (*Canella et al., 2014*; *Costa Louzada et al., 2015*; *Louzada et al., 2015a*; *Louzada et al., 2015b*; *Martinez Steele et al., 2016*; *Moubarac et al., 2013*). However, the relationship between the percent of energy intake from ultra-processed foods (PEI-UPF) during pregnancy and maternal and neonatal health outcomes has not been examined. Therefore, the purpose of this study was to determine the association between UPF consumption in pregnant US women and selected maternal/newborn health outcomes.

To do this, we used data collected by Tinius et al. on the health of 45 pregnant women and their neonates in St. Louis, MO, USA (*Tinius et al., 2015*; *Tinius et al., 2016*). In the original study's design, only women within the normal or obese BMI ranges (18.0–24.9 kg/m$^2$ or 30.0–45.0 kg/m$^2$) were included. Overweight women (BMI of 25.0–29.9 kg/m$^2$) were excluded. It was found that the lean and obese groups only differed in gestational weight gain and maternal weight. No significant differences in PEI-UPF or other clinical outcomes were found between the two groups. However, the two groups were modeled as having

different slopes (with respect to PEI-UPF) as well as intercepts to allow greater model flexibility.

We hypothesize that the percent of energy intake coming from UPF could serve as a concise measure of the diet quality of this sample of pregnant US women. Further, we hypothesize that PEI-UPF could be an efficient predictor of maternal and neonatal health outcomes. These include maternal gestational weight gain (GWG) and neonatal anthropometrics. The ability of UPF consumption to predict these outcomes is clinically important as high GWG is generally associated with high postpartum weight retention (*Gunderson & Abrams, 1999*), and with the child having a higher BMI early in life (*Lau et al., 2014*; *Mourtakos et al., 2016*). More broadly, research has shown that maternal obesity can negatively influence neonatal outcomes in a variety of ways (*Castro & Avina, 2002*). These patterns almost certainly do not end at birth: *Catalano et al. (2003)* found that infant body fat percentage in particular (as opposed to body weight) can be a significant predictor of early childhood, and possibly adult, obesity. Additionally, skinfold thickness measurements can be a predictor of insulin resistance and diabetes later in life (*Yajnik et al., 2003*). Therefore, the ability to determine the role of UPF consumption in maternal and neonatal health is important.

A secondary aim of the study was to compare the abilities of PEI-UPF and another dietary quality index, the Healthy Eating Index-2010 (HEI-2010), to predict maternal GWG and neonatal body composition. The HEI-2010 is a number ranging from 0 (worst) to 100 (best) that reflects the consumption of desirable macronutrients and food groups (fruits, vegetables, etc.), and avoidance of unhealthy foods (refined grains, sodium, and empty calories). The HEI-2010 measures diet quality according to the 2010 Dietary Guidelines for Americans (*Guenther et al., 2014*), and has been shown to have significant associations with biomarkers and clinical outcomes in gravid and non-gravid adults (*Reedy et al., 2014*; *Shapiro et al., 2016*). However, HEI-2010 has not been directly compared with PEI-UPF in this regard. The HEI-2010 is often computed using 24-hour food recalls or FFQs such as the US National Institutes of Health's Diet History Questionnaire II (DHQ II) (*National Cancer Institute (NCI), 2010*), in which subjects reported their consumption of various unprocessed, prepared, and packaged foods over the past month. Tinius et al. administered the DHQ II to participants, and found that macronutrient intake was largely similar between lean and obese study groups, although active obese women tended to consume more fat than inactive obese women (*Tinius et al., 2015*; *Tinius et al., 2016*). We note that the DHQ II can be used in a variety of other ways, such as measuring how many servings of a food were consumed (*Yang & Rose, 2014*), or calculating consumption of ultra-processed foods.

## METHODS

### Study design

This study used data collected by Tinius et al. as described above. Approval for this study was granted by the Institutional Review Board at Washington University (IRB ID: 201306109). Written informed consent was obtained from each participant. More information about

how maternal and neonatal outcomes were collected can be found elsewhere (*Tinius et al., 2015*; *Tinius et al., 2016*).

In the original study, all women had viable singleton pregnancies and no evidence of fetal abnormalities (both confirmed by ultrasound), and were recruited near the end of their second trimester. The majority of maternal health markers were measured during two visits, both of which occurred between 32 and 37 weeks gestation. Visit 1 occurred, on average, at 34 weeks, while Visit 2 occurred, on average, at 35 weeks. Maternal dietary indices were based on the 30 days preceding Visit 1, physical activity data were based on the week following Visit 1, and HDL (along with LDL) were measured at Visit 2. Neonatal measurements were obtained after delivery and before discharge from the hospital. In our study, key outcomes included maternal GWG and net triglyceride levels, as well as neonatal percent body fat and site-specific skinfold measurements. Free fatty acids, fasting insulin/glucose and C-reactive protein were measured in both mother and infant. These data were obtained as part of previously published studies (*Tinius et al., 2015*; *Tinius et al., 2016*).

## Survey instrument

As part of Visit 1, Tinius et al. administered the US National Institutes of Health's Diet History Questionnaire II (DHQ II) (*National Cancer Institute (NCI), 2010*). For the present study, the DHQ II was primarily used to calculate the percentage of energy intake that comes from ultra-processed foods (PEI-UPF). The HEI-2010, total energy intake, and total fat intake were also calculated to compare their predictive abilities, in terms of maternal and neonatal outcomes, with that of PEI-UPF. For each food on the DHQ II, the participant was asked to choose one of eight options that best characterized the frequency of consumption, ranging from "never" to "2 or more times per day". For beverages, options ranged from "never" to "6 or more times per day". Participants chose one of three options of typical serving sizes that best described the amount consumed. The total amount consumed per month was determined by multiplying the average of the frequency range with the average of the amount range. For condiments, participants chose one of five options reflecting what fraction of the time it was added to the main food. Dietary supplements were not considered.

The amount of each food consumed per month was converted to grams using a US Department of Agriculture (USDA) database. Each food was classified, according to the NOVA classification scheme, as (1) an unprocessed or minimally processed food, (2) a processed culinary ingredient, (3) a processed food, or (4) an ultra-processed food. Thirty-three subgroups (nested within the main groups) were used to further classify the foods. The quantities of seven different nutrients obtained from each group/subgroup were then calculated for each subject. Due to energy content inaccuracies in the USDA database, the energy in 100 g of each food had to be recalculated as follows:

$$\text{Energy (MJ)} = 0.017 \frac{\text{MJ}}{\text{gram}} \cdot \left(\text{Grams Carbohydrate} + \text{Grams Protein}\right)$$
$$+ 0.037 \frac{\text{MJ}}{\text{gram}} \cdot \text{Grams Fat}$$

where MJ represents megajoules.

In general, when several different foods (such as jam, jelly, and honey) were combined in a single question, nutrient information from the most commonly consumed food was used.

## Data management

Microsoft Excel 2013 was used for data entry, and spreadsheets were imported into R 3.2.3 (*R Core Team, 2015*) for calculations and statistical analysis. Several tables were automatically constructed using the stargazer package (*Hlavac, 2015*) within R. Missing frequency or amount data for individual foods were estimated using random forest imputation, through the missForest package in R (*Stekhoven, 2013*).

The HEI-2010 was computed using the Diet*Calc Analysis Program (*National Cancer Institute (NCI), 2012*) and the USDA's Food Patterns Equivalents Database. SAS version 9.4 (2002–2012; SAS Institute, Cary, NC, USA) was then used to run the National Cancer Institute's HEI-2010 scoring program.

## Statistical analysis

Simple matrix operations yielded the percentage of energy intake from ultra-processed foods (PEI-UPF) for each study participant. This number was used as the primary measure of diet quality. Diagnostic tests (for normality, linearity, independence, and homoscedasticity) were carried out to determine the appropriateness of linear modelling. Then, an ANCOVA-like model was used to analyze the relationship between PEI-UPF and the various clinical outcome variables.

For the analysis of maternal health outcomes, age (continuous), race (Caucasian or African American/other), weight status (lean or obese), socioeconomic status (Primarily Low-Income Clinic or Primarily High-Income Clinic), average daily energy and fat intake (continuous), and percent of time spent in moderate physical activity (continuous) were controlled for (Table 1). In the neonatal outcome analyses, we controlled for maternal age, race, weight status, socioeconomic status, average daily energy and fat intake, percent of time spent in moderate physical activity, and gestational age at which neonatal measurements were taken (continuous). All interactions with PEI-UPF were tested, and only significant interaction terms were included in the final models. However, the PEI-UPF * Obese Weight Status interaction was forced into all models, due to the special effect maternal obesity can have on neonatal outcomes. Essentially, the lean and obese groups each had a separate slope coefficient ($\beta$) for the effect of UPF consumption on the clinical outcome.

Extra sum-of-squares $F$-tests and adjusted $R^2$ values were used to compare the predictive ability of PEI-UPF and HEI-2010. Unlike $P$-values, which measure association, Adjusted $R^2$ measures the predictive power of a model, while correcting for the number of regressors (models with many extraneous regressors are penalized). Finally, since the assumption of normality was met, we used Pearson correlation to determine the association between HEI-2010 and PEI-UPF. All tests were two-sided, and $p < 0.05$ was considered significant.

**Table 1** Demographic and lifestyle characteristics of analyzed respondents, *n* = 45. Table 1 gives the frequencies for each level of relevant categorical variables, as well as mean and standard deviation for continuous variables.

| Maternal characteristics | Percentage |
| --- | --- |
| Race | |
|     Caucasian | 46.7% |
|     African-American | 46.7% |
|     Other | 6.7% |
| Clinic visited | |
|     Primarily low-income | 42.2% |
|     Primarily high-income | 57.8% |
| Parity | |
|     Nulliparous | 55.6% |
|     Multiparous | 44.4% |
| Weight status at beginning of study (i.e., before 32 weeks gestation) | |
|     Lean | 35.6% |
|     Obese | 64.4% |
| **Maternal characteristics (mostly at 32–37 weeks gestation)** | **Mean ± SD** |
| PEI-UPF (%) in the month preceding Visit 1 | 54.4 ± 13.2 |
| HEI-2010 (0–100) based on the month preceding Visit 1 | 62.2 ± 13.0 |
| Age (years) at Visit 1 | 27.2 ± 5.1 |
| Gestational age at Visit 1 (weeks) | 33.6 ± 1.4 |
| Gestational age at Visit 2 (weeks) | 34.7 ± 1.3 |
| Pre-pregnancy BMI at initiation of prenatal care | 30.1 ± 7.3 |
| Body fat (%) at Visit 1 | 31.8 ± 8.5 |
| Gestational weight gain (kg) between beginning of study and admission for labor/delivery | 12.0 ± 7.2 |
| HDL (mg/dL) at Visit 2 | 67.6 ± 15.3 |
| LDL (mg/dL) at Visit 2 | 121.4 ± 36.7 |
| Time spent in moderate physical activity (%) in the week following Visit 1 | 13.8 ± 4.1 |
| **Newborn characteristics (within 48 h of delivery)** | **Mean ± SD** |
| Gestational age when neonatal measurements taken | 39.6 ± 1.2 |
| Thigh skinfold thickness (mm) | 6.6 ± 1.4 |
| Subscapular skinfold thickness (mm) | 4.4 ± 0.8 |
| Body fat (%) | 11.5 ± 3.5 |

**Notes.**
Due to rounding, not all percentages may add to exactly 100.

# RESULTS

The present study is based upon previously published data with a sample size of *n* = 50. However, records with missing FFQ or clinical outcome data had to be excluded from this study. Of the final sample (*n* = 45), sixteen women are from the lean study group (*n* = 16) while the remainder (*n* = 29) are from the obese group. Detailed subject characteristics are presented in Table 1. The majority of women visited a primarily high-income clinic

**Table 2  Average nutrient intake by food group, $n = 45$.** Table 2 shows that a majority of energy intake (54.4%, on average) was obtained from ultra-processed foods, but at the same time processed foods represent a significant source of fat and sodium, and cannot be disregarded.

| Food groups | Mean intake | | | | | | |
|---|---|---|---|---|---|---|---|
| | Absolute (MJ/day) | Carbohydrate (% of total intake) | Protein (% of total intake) | Fat (% of total intake) | Total sugars (% of total intake) | Fiber (% of total intake) | Sodium (% of total intake) |
| 1. Unprocessed or minimally processed foods | 3.7 | 39.7 | 40.8 | 27.3 | 37.5 | 56.4 | 16.0 |
| 2. Processed culinary ingredients | 0.2 | 0.9 | 0.1 | 3.6 | 1.3 | 0 | 0.8 |
| 3. Processed foods | 0.8 | 2.4 | 22.6 | 10.3 | 3.3 | 3.7 | 17.6 |
| 4. Ultra-processed foods | 5.8 | 57.0 | 36.5 | 58.8 | 57.9 | 39.9 | 65.7 |
| TOTAL | 10.5 | 100 | 100 | 100 | 100 | 100 | 100 |

(57.8%), were nulliparous (55.6%), and obese (64.4%). Equal numbers of women were Caucasian and African American (46.7% each), and the remaining 6.7% were Hispanic or Asian. The average PEI-UPF was 54.4 ± 13.2% and the average percentage of energy intake for both processed and ultra-processed foods together was 63.2% (not shown in table). Among ultra-processed foods, the most consumed subgroup was Cakes, Cookies and Pies (5.8% of total energy). Only two out of all thirty-three subgroups had higher average consumption—fruits (9.1% of total energy intake) and grains (9.8%) (not shown in table).

Further detail showing the quantity of nutrients obtained from each main food group is given in Table 2. As with energy intake, the participants' total carbohydrate, fat, sugar and sodium intakes were primarily derived from ultra-processed foods (57.0%, 58.8%, 57.9% and 65.7% of total dietary intake, respectively). On the other hand, 39.9% of fiber was obtained from Group 4 foods. Indeed, pregnant women who limited their intake of ultra-processed foods tended to have better health outcomes for themselves and their infants. Tables 3 and 4 present the detailed results of multiple regression analysis on newborn and maternal outcomes, respectively. The association between PEI-UPF and GWG was observed only in the fully adjusted model, after controlling for maternal age, race, socioeconomic status, weight status, average daily energy and fat intake, and time spent in moderate physical activity. Likewise, the association of PEI-UPF with newborn body composition was observed only after controlling for maternal age, race, socioeconomic status, weight, average daily energy and fat intake, time spent by the woman in moderate physical activity, and gestational age at time of measurement. However, in each of the four models, the mother's weight status (lean or obese) had no significant slope or intercept effect on the relation between PEI-UPF and the clinical outcome. A number of biomarkers including blood levels of triglycerides (data available for mother only), free fatty acids, fasting glucose/insulin, and C-reactive protein had no significant association with PEI-UPF in either mothers or infants.

**Table 3  Associations between PEI-UPF and Gestational Weight Gain, adjusted for maternal characteristics, $n = 45$.** According to Table 3, PEI-UPF as well as the interaction between PEI-UPF and Age are significantly associated with GWG.

| Subject characteristic | Gestational weight gain (kg) | | |
|---|---|---|---|
| | $\beta$ | 95% CI | P-value |
| PEI-UPF (%) in the month preceding Visit 1 | 1.3 | (0.3, 2.4) | *0.016* |
| Age (years) at Visit 1 | 2.6 | (0.6, 4.6) | *0.014* |
| PEI-UPF * Age | −0.05 | (−0.09, −0.01) | *0.012* |
| Maternal weight status (ref: Lean) | | | |
| Obese | −5.1 | (−25.1, 15.0) | 0.61 |
| PEI-UPF * Obese | 0.06 | (−0.3, 0.4) | 0.72 |
| Avg. daily energy intake (kcal) | 0.003 | (−0.002, 0.008) | 0.20 |
| Avg. daily fat intake (g) | −0.06 | (−0.2, 0.07) | 0.38 |
| Race (ref: Caucasian) | | | |
| African-American/other | −7.9 | (−13.7, −2.2) | *0.0085* |
| Clinic visited (ref: primarily low-income) | | | |
| Primarily high-income | −2.0 | (−7.6, 3.6) | 0.47 |
| Time spent in moderate physical activity (%) in the week following Visit 1 | −0.2 | (−0.8, 0.5) | 0.58 |

**Notes.**
Gestational weight gain was measured from beginning of study until admission for labor/delivery.
Text in *italics* represents $P$-value $< 0.05$.

Various interaction terms with PEI-UPF were tested, and only the interaction with age was found to be significant ($p \leq 0.030$ for all four outcome variables). Thus, for older pregnant women, increased PEI-UPF has less of an effect on poor health outcomes than for younger women, as indicated by the negative coefficients for the interaction terms. All other interaction terms with PEI-UPF were not significant.

The predictive ability of PEI-UPF was also compared with several other measures. In adjusted models with only one dietary predictor (PEI-UPF, HEI-2010, total energy intake or total fat intake), PEI-UPF was more strongly associated with all clinical outcomes than either total energy or fat intake (Table 5). Indeed, PEI-UPF retained a significant relationship with GWG ($p = 0.016$) (Table 3), as well as neonatal thigh skinfold thickness ($p = 0.045$), subscapular skinfold thickness ($p = 0.026$) and body fat percentage ($p = 0.037$) (Table 4), even after controlling for total energy and fat intake. This suggests that PEI-UPF measures an aspect of diet that is independent of total energy and total fat intake.

Overall, PEI-UPF had a strong negative correlation with HEI-2010, with $r = -0.74$ 95% CI [−0.85, −0.56], indicating that both measures of diet quality are fairly consistent. Additionally, in models with one dietary predictor, maternal HEI-2010 scores were strongly associated with HDL cholesterol ($p = 0.0020$) (not shown in table), GWG ($p = 0.0011$), and neonatal subscapular skinfold thickness ($p = 0.026$) (Table 5).

In fully adjusted models including total energy and fat intake, the HEI-2010 was a better predictor of gestational weight gain than PEI-UPF (Adj. $R^2 = 0.26$, as opposed to 0.14 for PEI-UPF). However, PEI-UPF was still a better predictor of infant body fat percentage, thigh skinfold thickness, and subscapular skinfold thickness than the HEI-2010 (Adj. $R^2$

**Table 4   Associations between PEI-UPF and neonatal outcomes, adjusted for maternal characteristics, *n* = 45.** Table 4 shows that PEI-UPF as well as the interaction between PEI-UPF and Age are significantly associated with thigh skinfold thickness, subscapular skinfold thickness, and body fat percentage in the newborn.

| Subject characteristic | Newborn outcome (measured within 48 h of delivery) | | | | | | | | |
|---|---|---|---|---|---|---|---|---|---|
| | Thigh skinfold thickness (mm) | | | Subscap. skinfold thickness (mm) | | | Body fat (%) | | |
| | β | 95% CI | *P*-value | β | 95% CI | *P*-value | β | 95% CI | *P*-value |
| PEI-UPF (%) in the month preceding Visit 1 | 0.2 | (0.005, 0.4) | *0.045* | 0.1 | (0.02, 0.3) | *0.026* | 0.6 | (0.04, 1.2) | *0.037* |
| Age (years) at Visit 1 | 0.4 | (0.03, 0.8) | *0.035* | 0.3 | (0.06, 0.5) | *0.015* | 1.3 | (0.2, 2.4) | *0.023* |
| PEI-UPF * Age | −0.008 | (−0.02, −0.0008) | *0.030* | −0.006 | (−0.01, −0.001) | *0.014* | −0.02 | (−0.05, −0.004) | *0.020* |
| Maternal weight status (ref: Lean)—Obese | −2.6 | (−6.6, 1.4) | 0.19 | −0.8 | (−3.1, 1.4) | 0.46 | −3.0 | (−13.7, 7.7) | 0.58 |
| PEI-UPF * Obese | 0.06 | (−0.01, 0.1) | 0.098 | 0.02 | (−0.02, 0.06) | 0.35 | 0.09 | (−0.1, 0.3) | 0.35 |
| Maternal Avg. daily energy intake (kcal) | −0.0009 | (−0.002, 0.0001) | 0.081 | 0.0002 | (−0.0004, 0.0007) | 0.55 | 0.0009 | (−0.002, 0.004) | 0.48 |
| Maternal Avg. daily fat intake (g) | 0.03 | (0.003, 0.06) | *0.030* | −0.0008 | (−0.02, 0.01) | 0.91 | −0.01 | (−0.08, 0.06) | 0.70 |
| Race (ref: Caucasian)—African-American/other | −0.3 | (−1.4, 0.9) | 0.62 | −0.2 | (−0.8, 0.5) | 0.63 | 0.3 | (−2.7, 3.4) | 0.83 |
| Clinic visited (ref: primarily low-income)—primarily high-income | 0.3 | (−0.8, 1.5) | 0.57 | −0.08 | (−0.7, 0.6) | 0.81 | 1.4 | (−1.7, 4.6) | 0.36 |
| Gestational age when neonatal measurements taken (weeks) | 0.3 | (−0.05, 0.7) | 0.082 | 0.2 | (0.01, 0.5) | *0.041* | −0.1 | (−1.2, 1.0) | 0.83 |
| Time spent in moderate physical activity (%) in the week following Visit 1 | −0.05 | (−0.2, 0.08) | 0.45 | −0.004 | (−0.08, 0.07) | 0.91 | 0.04 | (−0.3, 0.4) | 0.83 |

**Notes.**
Text in *italics* represents *P*-value < 0.05.

= 0.01, 0.14, and 0.10, as opposed to −0.09, −0.02, and −0.02, respectively) (not shown in table). Although HEI-2010 has a greater association with subscapular skinfold thickness than PEI-UPF (according to *P*-values in Table 5), Adjusted $R^2$ values indicate that overall the HEI-2010 model is a worse predictor than the PEI-UPF model. Furthermore, adding HEI-2010 as a predictor in our four fully adjusted PEI-UPF models did not significantly improve fit ($p \geq 0.097$ from extra sum-of-squares $F$-test in all cases). The failure of HEI-2010 to improve model fit was likely caused by the strong (negative) correlation between PEI-UPF and HEI-2010.

## DISCUSSION

The results show a strong positive association of PEI-UPF with GWG and with neonatal anthropometrics (i.e., subscapularis and thigh skinfold thicknesses and body fat percentage). This study demonstrates that many pregnant women are obtaining the majority of their energy from ultra-processed foods, and these ultra-processed foods may

**Table 5** *P*-values for various dietary indices in models with only one dietary index. Table 5 shows that for most of the clinical outcomes, PEI-UPF is a significant predictor even in the absence of other dietary predictors. HEI-2010 is sometimes a significant predictor, but Total Energy Intake and Total Fat Intake are not significant for any of the outcomes tested.

| Dietary index | Maternal or newborn outcome | | | |
|---|---|---|---|---|
| | Gestational weight gain (kg) | Thigh skinfold thickness (mm) | Subscap. skinfold thickness (mm) | Body fat (%) |
| PEI-UPF | *0.017* | 0.12 | *0.036* | *0.035* |
| HEI-2010 | *0.0011* | 0.41 | *0.026* | 0.30 |
| Total energy intake | 0.73 | 0.45 | 0.80 | 0.97 |
| Total fat intake | 0.88 | 0.59 | 0.75 | 0.76 |

**Notes.**
All models were adjusted for age (continuous), race (Caucasian or African American/other), weight status (lean or obese), socioeconomic status (Primarily Low-Income Clinic or Primarily High-Income Clinic), and percent of time spent in moderate physical activity (continuous). Models for newborn outcomes were also adjusted for gestational age at which neonatal measurements were taken (continuous).
Text in *italics* represents *P*-value < 0.05.

also be worsening health outcomes for themselves and their children. These relationships are essentially the same in both lean and obese mothers. Indeed, the majority of participants' carbohydrate, fat, sugar, sodium, and energy were obtained from UPF, which is consistent with the refined ingredients and highly palatable nature of such foods. As such, it is not surprising that UPF consumption negatively affects health.

The identification of causes of excessive gestational weight gain is clinically important as excessive gestational weight gain can have serious consequences for the postpartum women and their neonates. It leads to excessive postpartum weight retention (*Gunderson & Abrams, 1999*), which in turn can contribute to long-term obesity and associated comorbidities including type 2 diabetes, cardiovascular disease, mental health issues, and cancer (*Institute of Medicine and National Research Council, 2009*). For the neonate, excess adiposity is likely to continue into childhood (*Mei, Grummer-Strawn & Scanlon, 2003*), and childhood obesity is a strong predictor of adult obesity (*Freedman et al., 2005*). Thus, higher body fat as an infant may contribute to long-term risk for obesity and its associated comorbidities (*Catalano & Ehrenberg, 2006*; *Tinius, Cahill & Cade, 2016*). Because UPF consumption was related not only to excessive gestational weight gain, but also neonatal adiposity, maternal diet quality modification could substantially improve long-term health outcomes for mother and child.

Interestingly, a number of successful interventions to limit excessive GWG emphasize energy or fat restriction, or other macronutrient targets (*Gardner et al., 2011*; *Phelan et al., 2011*). Despite the popularity of energy- and fat-restricting diets, GWG was more strongly associated with PEI-UPF than total energy or fat intake ($p = 0.017$ for PEI-UPF compared to $p = 0.73$ and $p = 0.88$ for total energy and fat intake). More generally, although low fat ultra-processed foods are ubiquitous, such results cast doubt on the health benefits of these low fat foods. We believe interventions to limit GWG could be even more successful if they also emphasized a minimally-processed diet, since our results show that PEI-UPF captures information about diet quality, which total fat or energy intake cannot. In general, our results suggest that consumption of UPF may be a key factor

contributing to unfavorable maternal and neonatal outcomes. This study showed that poor diet quality during pregnancy increases neonatal adiposity independent of maternal weight and maternal moderate physical activity; thus, maternal diet quality is an important direction of future study. Specifically, diet quality seems to be more important than the amount of energy consumed. Thus, from a clinical standpoint, pregnant women should be educated to focus less on the total energy consumed, and more on the source of that energy.

Interestingly, the nutrient profiles indicate that processed foods generally have less fiber and more sodium than UPF. Thus, while small amounts of processed foods are part of any diet and acceptable in moderation, clear preference should be given to unprocessed/minimally processed foods. It is also important to highlight the need for moderation when using salt, sugar and oil in home based meal preparations.

Furthermore, almost all currently used tools for assessing diet are largely quantity based instead of quality based, which is one of the main reasons for measuring UPF consumption. Because many currently used tools assess quantity, we also wanted to assess the ability of the HEI-2010 to predict maternal and neonatal outcomes. As part of our secondary aim, our results did show that a quantity-focused measure such as HEI-2010 can be a useful predictor of gestational weight gain. Despite the high correlation between PEI-UPF and HEI-2010, PEI-UPF is a better predictor of neonatal body fat percentage and skinfold thickness at the thigh and subscapularis. This comparison is based on Adjusted $R^2$ values (0.01, 0.14, and 0.10 for PEI-UPF, as opposed to $-0.09$, $-0.02$, and $-0.02$ for HEI-2010). Interestingly, *Shapiro et al. (2016)* found that a low maternal HEI-2010 score was associated with higher neonatal body fat percentage, in a sample size of >1,000 woman and infant pairs. We were unable to confirm this finding (in our study, $p = 0.30$ for association with body fat percentage). The differences in predictive ability between HEI-2010 and PEI-UPF indicate that each statistic measures different aspects of the diet, and therefore both are useful. To achieve the optimal diet, one must both limit intake of UPFs as well as eat a variety of different nutrients.

This study has several notable strengths, including being the first effort to measure UPF consumption in pregnant women, and to correlate PEI-UPF with maternal and neonatal clinical outcomes. Additionally, it is only the second study to examine PEI-UPF in the United States, where the percent of the diet coming from UPFs is much higher than in some other countries (*Canella et al., 2014*; *Martinez Steele et al., 2016*; *Monteiro et al., 2013*). However, this study presents some limitations. Due to the design of the original longitudinal study, only women within the normal or obese weight ranges were included (BMI between 18.0 kg/m$^2$ and 24.9 kg/m$^2$ or between 30.0 kg/m$^2$ or 45.0 kg/m$^2$). Thus, women in the overweight range were excluded, and the study results may not be applicable to such women. Additionally, the racial composition, with essentially equal numbers of Caucasians and African Americans, and very few other minorities, is not representative of the entire US population. The design of the survey instrument presents further limitations. Since food frequency and portion sizes were collected in a semi-quantitative/categorical format, often with as few as three options, there was some error simply because respondents had to round off quantities. Additionally, somewhat subjective researcher input was

required to categorize each DHQ II food according to the NOVA scheme. For example, homemade bread would be a processed food, but for this study, bread was classified as ultra-processed since most bread consumed in the US meets this definition. A full listing of classifications (along with justification) can be found in the Appendix S1.

A greater error we could not eliminate is the fact that participants may underreport their food intake. Previous research found that postmenopausal women underestimated their energy intake by 21% on a FFQ (*Horner et al., 2002*). However, another study found that food frequency questionnaires (FFQs) inquiring about consumption over a several-month period provide reproducible and valid measures of relative dietary intakes in pregnant populations (*Vioque et al., 2013*). However, since we are using percentages of energy intake as the main predictor rather than absolute energy, we feel our data may not be subject to the same degree of error. Finally, another major limitation is that administering the DHQ II once, at Visit 1, effectively only assesses maternal diet at 30–34 weeks gestation. It is unlikely that this assessed diet accurately represents diet across the entire pregnancy, since previous research indicates that intake of certain foods and overall caloric intake vary across the three trimesters (*Durnin, 1991*; *Rifas-Shiman et al., 2006*).

## CONCLUSIONS

This study showed that consumption of ultra-processed foods leads to unfavorable pregnancy outcomes including excessive maternal gestational weight gain and increased neonatal body fatness. For both mother and neonate, excess adiposity is likely to remain, contributing to associated comorbidities such as Type II diabetes, cardiovascular disease, mental health issues, cancer. Reducing dietary consumption of ultra-processed foods may be a potential avenue for improving short and long term maternal and neonatal health, making this an important direction for future research. A natural, minimally-processed diet centered on home cooking should be promoted among pregnant women.

## ACKNOWLEDGEMENTS

We are grateful to Carlos A. Monteiro from University of Sao Paulo for his valuable input regarding the NOVA classification system and analyses. We are also grateful to Mary Brugge and Nicholas Schroeder, DPT (Doctorate in Physical Therapy) students at Washington University in St. Louis, for their help with preparation of the data for analysis. We wish to thank Ken Bishop for his assistance with quality control in our DHQ II data file. Special thanks to the study participants and their newborns.

### Funding

This research was funded by the National Institutes of Health grants TL1 TR000449, DK094416, Nutrition Obesity Research Center Grant DK56341, Diabetes Research Center Grant DK20579, and Clinical and Translational Science Award RR024992. There was no

additional external funding received for this study. The funders had no role in study design, data collection and analysis, decision to publish, or preparation of the manuscript.

## Grant Disclosures

The following grant information was disclosed by the authors:
National Institutes of Health: TL1 TR000449, DK094416.
Nutrition Obesity Research Center: DK5634.
Diabetes Research Center: DK20579.
Clinical and Translational Science Award: RR024992.

## Competing Interests

Diana C. Parra is an Academic Editor for PeerJ.

## Author Contributions

- Karthik W. Rohatgi performed the experiments, analyzed the data, wrote the paper, prepared figures and/or tables.
- Rachel A. Tinius performed the experiments, wrote the paper, reviewed drafts of the paper.
- W. Todd Cade performed the experiments, reviewed drafts of the paper.
- Euridice Martínez Steele analyzed the data, reviewed drafts of the paper.
- Alison G. Cahill conceived and designed the experiments, performed the experiments, reviewed drafts of the paper.
- Diana C. Parra conceived and designed the experiments, analyzed the data, wrote the paper, reviewed drafts of the paper.

## Human Ethics

The following information was supplied relating to ethical approvals (i.e., approving body and any reference numbers):

The Institutional Review Board at Washington University in St. Louis granted approval to carry out the study (IRB ID: 201306109).

## Data Availability

The raw data and R code have been provided as Supplemental Files.

## Supplemental Information

Supplemental information for this article can be found online at http://dx.doi.org/10.7717/peerj.4091#supplemental-information.

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
