# Peer review of "Relationships between consumption of ultra-processed foods, gestational weight gain and neonatal outcomes in a sample of US pregnant women"

_PeerJ, doi:10.7717/peerj.4091_

## Round 0.1 · original submission · Minor Revisions

· Academic Editor

Minor Revisions

The manuscript has been reviewed by two experts in the field and they have raised minor issues that needs to be addressed prior for further consideration of the manuscript

Reviewer 1 ·

Basic reporting

The present study is a longitudinal study aimed to evaluate the relationship between percent of energy intake from ultra-processed foods during pregnancy and maternal and neonatal outcomes related to weight gain and body composition.
The article adheres to PeerJ policies in terms of structure, and is written in a clear language providing eloquent explanations.
The introduction section is clear and outlines basic principles about dietary intake during pregnancy that aid in understanding the study design and results sections. However there are certain aspects not discussed in the introduction – for example the outcomes of previously published articles using the same data, or the use of the Diet History questionnaire II. As a result these are discussed in the methods section which becomes over-detailed (see examples at “Experimental design” below).

Specific comments regarding introduction:
• In lines 33-34 “… has been recommended” – a specific reference should be related to these recommendations. If the authors are referring to recommendations by Imhoff-Kunsch & Martorell 2012 then the location of the reference (now appears as if relates only to previous sentence) should be changed accordingly.
• Line 64 – “critical” is a strong word to describe the role of UPF consumption in maternal and neonatal health. Maybe “important” or “of importance” or any other choice of words should be considered.

Figures and tables are relevant to the content of the article, and are adequately labeled. However, sometimes the text lacks adequate referral to the tables. The first time that tables 3+4 are presented in the text (line 194) there should be a clear referral as to what information the table holds (like previously reported for table 1+2), so that the even a reader that does not want to approach the table at the same time will understand it’s value. When citing data from table 5 in lines 220-221 it is advisable to include p-values in the text since these are the major outcomes of the study (same as should be mentioned for all outcomes in the text).
Specific comments –
• The use of an asterisk in the tables to emphesize p-value<0.005 is confusing in tables 3&4 when it is also used in the first column (“for PEI-UPF * Age” etc.). Alternative symbol can be used or significant p-values can be bold text.

Experimental design

The study design relies on data collected for previously published studies by the same group. This leads to some undefined issues in the study design.

The choice of two groups of patients (lean & obese) and the exclusion of overweight patients is indeed cited as a limitation on the discussion section; however it does require an adequate explanation in the “study design” section. Lines 92-101 describe a comparison between these two groups – if it is a previous one it should be in the introduction and shortened, if this is part of the present study – it should be incorporated into the “results” section, and be accompanied by the p-values and possibly a table (a revision of table 1 to show also subgroup specific data and p-values).
Several background explanations should be edited out of the methods study and incorporated either in introduction or in discussion, as appropriate. For example:
• Lines 105-114 from “Previous research has shown…” to “such as the questionnaire used in the present study”. Or at least until line 109.
• Lines 97-99 – explanation of prior research should be in introduction unless a specific method cited in this article is utilized in current study. Lines 116-124 – a little too lengthy, should consider referring to appendix.

The comparison of PEI-UPS to total fat intake and total energy intake is not mentioned in the methods, although is discussed in results section and mentioned in the abstract under “Methods” (“The ability of these dietary indices to predict gestational weight gain was also compared with the predictive abilities of total energy intake and total fat intake”).

Specific comments:
• Lines 84-87 – the authors do not address the difference in the 2 prenatal visits, and therefore the reader cannot assess the importance of the fact that on average there was no difference in gestational age. If, for example this is relevant for GWG assessment than some data about the average interval between visits could be more appropriate.
• Lines 158-161 – exactly repeat lines 87-89 without adding information relevant to the statistical analysis of these outcome measures.

Validity of the findings

“Results” section -
Even though this is an analysis of pre-published data the results section should be wholesome and represent all information regarding results. Number of patients recruited, how many in each group (lean/obese) etc. prior to presenting table 1.
Specific comments –
• All data should be presented with p-value in the text, even if p-value appears in table (example line 217-218, 220-221, 223)

“Discussion” section –
The comparison between HEI-2010 and PEI-UPP discussed in lines 275-284 is unclear and partially inconsistent with data presented in table 5. Lines 277-278 state that PEI-UPF (note spelling mistake since this line says UPP) is a better predictor of neonatal body fat percentage which is consistent with the table (although requires p-values in the text and referral to table). However according to table 5 for skinfold thickness at the subscapularis both tests are significant (and HEI-2010 even with a lower p-value!), and for the thigh both are non-significant. Therefore when stated in live 278 that PEI-UPF is a better predictor in all these categories it is not clear on which statistical parameter the authors rely for this statement (they do not refer to any tale or p-value at this section). Additionally in line 281 the p value stated for association of HEI-2010 with body fat percentage is stated to be 0.334 whereas in table 5 it is 0.3 and the value of 0.334 cannot be found elsewhere in the data.
Line 303 – not only pregnant women may underreport food intake, it is a human tendency. The words “subjects” or “participants” is probably more appropriate.

“Conclusions” section –
The conclusion sections should briefly outline the main findings of the study without repeating general observations already presented in the discussion. Therefore should consider omitting lines 315-317 (from “excessive gestational weight gain”…”to “associated comorbidities”) and lines 319-320 (from “thus, from a clinical standpoint…” to “that energy”).

Additional comments

This study is addressing important issues regarding maternal nutrition during pregnancy and its effect on maternal gain weight and neonatal obesity measures. This field is of growing interest in many countries.

·

Basic reporting

Manuscript : 18220v1
Title” Relationships between consumption of ultra-processed foods, gestational weight gain and neonatal outcomes in a sample of US pregnant women”
1. Title: It is clear.
2. Abstract : Background, Methods, Results are Ok .
In "Discussion":
1) Spell out "PEI-UPF".
2) The first sentence should be more specific.
3) Instead of 'several maternal outcomes", the authors should say "increase gestational weight gain" .
4) Instead of " several neonatal clinical outcomes", the authors should say" increase neonatal body fatness"
3. Introduction: It is clear and to the point.
4. Method : It is very well written and clear.
5. Results and Discussion are very well written
6. Conclusions: It should shortened. I suggest to delete the two sentences on line 315 and 319 that start with “Excessive gestational weight….” and ends with “ food consumed.”.
7. References are pertinent and updated.
8. Tables and Appendix are OK.

Experimental design

"No comments"

Validity of the findings

The results are novel and important.

Additional comments

Important article that demonstrates that diet quality seems to be more important than the amount of food consumed in pregnancy.

---

## Round 0.2 · accepted · Accept

· Academic Editor

Accept

The manuscript has received favorable reviews and the reviewers comments were well addressed, therefore this manuscript is accepted for publication.

Reviewer 1 ·

Basic reporting

NA

Experimental design

NA

Validity of the findings

NA

Additional comments

The authors have revised the manuscript in accordance to our previous comments, and is now vastly improved.